# BETTER FINE-TUNING BY REDUCING REPRESENTATIONAL COLLAPSE

**Armen Aghajanyan, Akshat Shrivastava, Anchit Gupta & Naman Goyal**
Facebook
{armenag,akshats,anchit,naman}@fb.com

**Luke Zettlemoyer & Sonal Gupta**
Facebook
{lsz, sonalgupta}@fb.com

## ABSTRACT

Although widely adopted, existing approaches for fine-tuning pre-trained language models have been shown to be unstable across hyper-parameter settings, motivating recent work on trust region methods. This paper presents a simplified and efficient method rooted in trust region theory that replaces previously used adversarial objectives with parametric noise (sampling from either a normal or uniform distribution), thereby discouraging representation change during fine-tuning when possible without hurting performance. We also introduce a new analysis to motivate the use of trust region methods more generally, by studying representational collapse; the degradation of generalizable representations from pre-trained models as they are fine-tuned for a specific end task. Extensive experiments show that our fine-tuning method matches or exceeds the performance of previous trust region methods on a range of understanding and generation tasks (including DailyMail/CNN, Gigaword, Reddit TIFU, and the GLUE benchmark), while also being much faster. We also show that it is less prone to representation collapse; the pre-trained models maintain more generalizable representations every time they are fine-tuned.

## 1 INTRODUCTION

Pre-trained language models (Radford et al., 2019; Devlin et al., 2018; Liu et al., 2019; Lewis et al., 2019; 2020) have been shown to capture a wide array of semantic, syntactic, and world knowledge (Clark et al., 2019), and provide the defacto initialization for modeling most existing NLP tasks. However, fine-tuning them for each task is a highly unstable process, with many hyperparameter settings producing failed fine-tuning runs, unstable results (considerable variation between random seeds), over-fitting, and other unwanted consequences (Zhang et al., 2020; Dodge et al., 2020).

Recently, trust region or adversarial based approaches, including SMART (Jiang et al., 2019) and FreeLB (Zhu et al., 2019), have been shown to increase the stability and accuracy of fine-tuning by adding additional constraints limiting how much the fine-tuning changes the initial parameters. However, these methods are significantly more computationally and memory intensive than the more commonly adopted simple-gradient-based approaches.

This paper presents a lightweight fine-tuning strategy that matches or improves performance relative to SMART and FreeLB while needing just a fraction of the computational and memory overhead and no additional backward passes. Our approach is motivated by trust region theory while also reducing to simply regularizing the model relative to parametric noise applied to the original pre-trained representations. We show uniformly better performance, setting a new state of the art for RoBERTa fine-tuning on GLUE and reaching state of the art on XNLI using no novel pre-training approaches (Liu et al., 2019; Wang et al., 2018; Conneau et al., 2018). Furthermore, the low overhead of our family of fine-tuning methods allows our method to be applied to generation tasks where we consistently outperform standard fine-tuning, setting state of the art on summarization tasks.

We also introduce a new analysis to motivate the use of trust-region-style methods more generally, by defining a new notion of representational collapse and introducing a new methodology for measuring it during fine-tuning. Representational collapse is **the degradation of generalizable representations of pre-trained models during the fine-tuning stage**. We empirically show that standard fine-tuning degrades generalizable representations through a series of probing experiments on GLUE tasks. Furthermore, we attribute this phenomenon to using standard gradient descent algorithms for the fine-tuning stage. We also find that (1) recently proposed fine-tuning methods rooted in trust region, i.e., SMART, can alleviate representation collapse, and (2) our methods alleviate representational collapse to an even greater degree, manifesting in better performance across almost all datasets and models.

Our contributions in this paper are the following.

- We propose a novel approach to fine-tuning rooted in trust-region theory, which we show directly alleviates representational collapse at a fraction of the cost of other recently proposed fine-tuning methods.

- Through extensive experimentation, we show that our method outperforms standard fine-tuning methodology following recently proposed best practices from Zhang et al. (2020). We improve various SOTA models from sentence prediction to summarization, from monolingual to cross-lingual.

- We further define and explore the phenomena of representational collapse in fine-tuning and directly correlate it with generalization in tasks of interest.

## 2 Learning Robust Representations through Regularized Fine-tuning

We are interested in deriving methods for fine-tuning representations that provide guarantees on the movement of representations, in the sense that they do not forget the original pre-trained representations when they are fine-tuned for new tasks (see Section 4 for more details). We introduce a new fine-tuning method rooted in an approximation to trust region, which provides guarantees for stochastic gradient descent algorithms by bounding some divergence between model at update $t$ and $t + 1$ (Pascanu & Bengio, 2013; Schulman et al., 2015b; Jiang et al., 2019).

Let $f : \mathbb{R}^{m \times n} \to \mathbb{R}^p$ be a function which returns some pre-trained representation parameterized by $\theta_f$ from $m$ tokens embedded into a fixed vector of size $n$. Let the learned classification head $g : \mathbb{R}^p \to \mathbb{R}^q$ be a function which takes an input from $f$ and outputs a valid probability distribution parameterized by $\theta_g$ in $q$ dimensions and let $X$ be our dataset. In the case of generation, we can assume the classification head is simply an identity function or softmax depending on the loss function. Let $\mathcal{L}(\theta)$ denote a loss function given by $\theta = [\theta_f, \theta_g]$.

We are interested in minimizing $\mathcal{L}$ with respect to $\theta$ such that each update step is constrained by movement in the representational density space $p(f)$. More formally given an arbitrary $\epsilon$

$$
\begin{aligned}
&\arg\min_{\Delta\theta} \mathcal{L}(\theta + \Delta\theta) \\
&\quad s.t. \ KL(p(f(\cdot\,;\theta_f))||p(f(\cdot\,;\theta_f + \Delta\theta_f))) = \epsilon
\end{aligned}
\tag{1}
$$

This constrained optimization problem is equivalent to doing natural gradient descent directly over the representations (Pascanu & Bengio, 2013). Unfortunately, we do not have direct access to the density of representations; therefore, it is not trivial to directly bound this quantity. Instead, we propose to do natural gradient over $g \cdot f$ with an additional constraint that $g$ is at most 1-Lipschitz (which naturally constrains change of representations, see Section A.1 in the Appendix). Traditional computation of natural gradient is computationally prohibitive due to the need for inverting the Hessian. An alternative formulation of natural gradient can be stated through mirror descent, using Bregmann divergences (Raskutti & Mukherjee, 2015; Jiang et al., 2019).

This method primarily serves as a robust regularizer by preventing large updates in the model's probability space. This family of methods is classically known as trust-region methods (Pascanu & Bengio, 2013; Schulman et al., 2015a).

$$\mathcal{L}_{SMART}(\theta, f, g) = \mathcal{L}(\theta) + \lambda \mathbb{E}_{x \sim X} \left[ \sup_{x^{\sim}:|x^{\sim}-x| \leq \epsilon} KL_S(g \cdot f(x) \| g \cdot f(x^{\sim})) \right] \quad (2)$$

However, the supremum is computationally intractable. An approximation is possible by doing gradient ascent steps, similar to finding adversarial examples. This was first proposed by SMART with a symmetrical $KL_S(X, Y) = KL(X\|Y) + KL(Y\|X)$ term (Jiang et al., 2019).

We propose an even simpler approximation which does not require extra backward computations and empirically works as well as or better than SMART. We altogether remove the adversarial nature from SMART and instead optimize for a smoothness parameterized by $KL_S$. Furthermore, we optionally also add a constraint on the smoothness of $g$ by making it at most 1-Lipschitz, the intuition being if we can bound the volume of change in $g$ we can more effectively bound $f$.

$$\mathcal{L}_{R3}(f, g, \theta) = \mathcal{L}(\theta) + \lambda \mathbb{E}_{x \sim X} \left[ KL_S(g \cdot f(x) \| g \cdot f(x+z)) \right] \qquad \textbf{R3F Method} \quad (3)$$
$$s.t. \quad z \sim \mathcal{N}(0, \sigma^2 I) \text{ or } z \sim \mathcal{U}(-\sigma, \sigma) \qquad (4)$$
$$s.t. \quad Lip\{g\} \leq 1 \qquad \qquad \text{Optional } \textbf{R4F Method} \quad (5)$$

where $KL_S$ is the symmetric KL divergence and $z$ is a sample from a parametric distribution. In our work we test against two distributions, normal and uniform centered around $0$. We denote this as the **R**obust **R**epresentations through **R**egularized **F**inetuning (**R3F**) method.

Additionally we propose an extension to R3F (**R4F**; **R**obust **R**epresentations through **R**egularized and **R**eparameterized **F**inetuning, which reparameterizes $g$ to be at most 1-Lipschitz via Spectral Normalization (Miyato et al., 2018). By constraining $g$ to be at most 1-Lipschitz, we can more directly bound the change in representation (Appendix Section A.1). Specifically we scale all the weight matrices of $g$ by the inverse of their largest singular values $W_{SN} := W/\sigma(W)$. Given that spectral radius $\sigma(W_{SN}) = 1$ we can bound $Lip\{g\} \leq 1$. In the case of generation, $g$ does not have any weights therefore we can only apply the R3F method.

## 2.1 RELATIONSHIP TO SMART AND FREELB

Our method is most closely related to the SMART algorithm, which utilizes an auxiliary smoothness inducing regularization term, which directly optimizes the Bregmann divergence mentioned above in Equation 2 (Jiang et al., 2019).

SMART solves the supremum by using an adversarial methodology to ascent to the largest KL divergence with an $\epsilon$−ball. We instead propose to remove the ascent step completely, optionally fixing the smoothness of the classification head $g$. This completely removes SMART's adversarial nature and is more akin to optimizing the smoothness of $g \cdot f$ directly. Another recently proposed adversarial method for fine-tuning, FreeLB optimizes a direct adversarial loss $\mathcal{L}_{FreeLB}(\theta) = \sup_{\Delta\theta:|\Delta\theta| \leq \epsilon} \mathcal{L}(\theta + \Delta\theta)$ through iterative gradient ascent steps. This is similar to SMART in the sense that both are adversarial and require gradient ascent steps. Unfortunately, the need for extra forward-backward passes can be prohibitively expensive when fine-tuning large pre-trained models (Zhu et al., 2019).

|  | FP | BP | xFP |
|---|---|---|---|
| FreeLB | $1 + S$ | $1 + S$ | $3 + 3S$ |
| SMART | $1 + S$ | $1 + S$ | $3 + 3S$ |
| R3F/R4F | 2 | 1 | 4 |
| Standard | 1 | 1 | 3 |

Table 1: Computational cost of recently proposed fine-tuning algorithms. We show Forward Passes (FP), Backward Passes (BP) as well as computation cost as a factor of forward passes (xFP). $S$ is the number of gradient ascent steps, with a minimum of $S \geq 1$

Our method is significantly more computationally efficient than adversarial based fine-tuning methods, as seen in Table 1. We show that this efficiency does not hurt performance; we can match or exceed FreeLB and SMART on a large number of tasks. In addition, the relatively low costs of our methods allow us to improve over fine-tuning on an array of generation tasks.

## 3 EXPERIMENTS

We will first measure performance by fine-tuning on a range of tasks and languages. The next sections report why methods rooted in trust region, including ours, outperform standard fine-tuning. We aimed for fair comparisons throughout all of our experiments by using fixed budget hyper-parameters searches across all methods. Furthermore, for computationally tractable tasks, we report median/max numbers as well as show distributions across a large number of runs.

### 3.1 SENTENCE PREDICTION

#### GLUE

We will first test R3F and R4F on sentence classification tasks from the GLUE benchmark (Wang et al., 2018). We select the same subset of GLUE tasks that have been reported by prior work in this space (Jiang et al., 2019): MNLI (Williams et al., 2018), QQP (Iyer et al., 2017), RTE (Bentivogli et al., 2009), QNLI (Rajpurkar et al., 2016), MRPC (Dolan & Brockett, 2005), CoLA (Warstadt et al., 2018), SST-2 (Socher et al., 2013).[1]

Consistent with prior work (Jiang et al., 2019; Zhu et al., 2019), we focus on improving the performance of RoBERTa-Large based models in the single-task setting (Liu et al., 2019). We report the performance of all models on the GLUE development set.

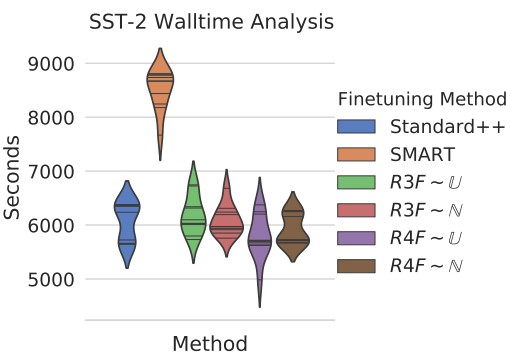

Figure 1: Empirical evidence towards the computational benefits of our method we present training wall time analysis on the SST-2 dataset. Each method includes a violin plot for 10 random runs. We define wall-time as the training time in seconds to best checkpoint.

We fine-tune each of the GLUE tasks with four methods: Standard (STD), the traditional fine-tuning scheme as done by RoBERTa (Liu et al., 2019); Standard++ (STD++), a variant of standard fine-tuning that incorporates recently proposed best practices for fine-tuning, specifically longer fine-tuning and using bias correction in Adam (Zhang et al., 2020); and our proposed methods R3F and R4F. We compare against the numbers reported by SMART, FreeLB, and RoBERTa on the validation set. For each method, we applied a hyper-parameter search with equivalent fixed budgets per method. Fine-tuning each task has task-specific hyper-parameters described in the Appendix (Section A.2). After finding the best hyperparameters, we replicated experiments with optimal parameters across ten different random seeds. Our numbers reported are the maximum of 10 seeds to be comparable with other benchmarks in Table 2.

In addition to showing the best performance, we also show the distribution of various methods across ten seeds to demonstrate the stability properties of individual methods in Figure 2.

R3F and R4F unanimously improve over Standard and Standard++ fine-tuning. Furthermore, our methods match or exceed adversarial methods such as SMART/FreeLB at a fraction of the computational cost when comparing median runs. We show computational cost in Figure 1 for a single task, but the relative behavior of wall times is consistent across all other GLUE tasks. We note that we could not find a discernable difference in the experimental setting, which would make the selection between R3F vs. R4F trivial.

---

[1]We do not test against STS-B because it is a regression task where our KL divergence is not defined (Cer et al., 2017).

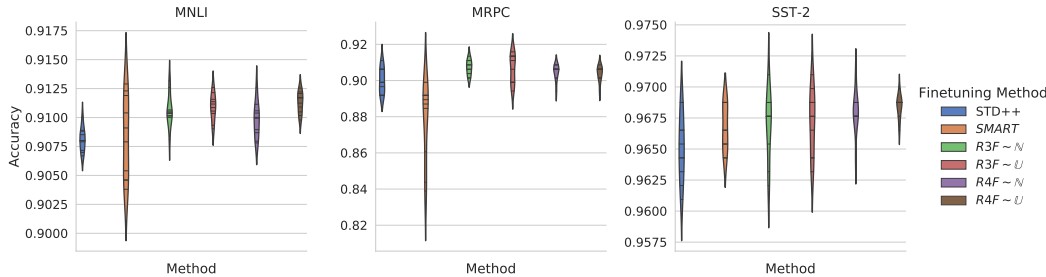

Figure 2: We show the results of our method against Standard++ fine-tuning and SMART across 3 tasks. Across 10 random seeds both *max* and *median* of our runs were higher using our method than both SMART and Standard++.

|  | MNLI Acc-m/mm | QQP Acc/F1 | RTE Acc | QNLI Acc | MRPC Acc | CoLA Mcc | SST-2 Acc | MNLI Acc-m/mm | QQP Acc/F1 | RTE Acc | QNLI Acc | MRPC Acc | CoLA Mcc | SST-2 Acc |
|---|---|---|---|---|---|---|---|---|---|---|---|---|---|---|
| STD | 90.2/- | 92.2/- | 86.6 | 94.7 | 89.1 | 68.0 | 96.4 | 90.2/- | 91.9/- | 86.6 | 92.1 | 84.4 | 66.2 | 96.4 |
| STD++ | 91.0/- | 92.2/- | 87.4 | 94.8 | 91.1 | 69.4 | 96.9 | 90.8/- | 92.1/- | 87.4 | 92.5 | 89.1 | 68.4 | 96.9 |
| FreeLB | 90.6/- | **92.6**/- | 88.1 | 95.0 | - | 71.1 | 96.7 | -/- | -/- | - | - | - | - | - |
| SMART | **91.1/91.3** | 92.4/89.8 | **92.0** | **95.6** | 89.2 | 70.6 | 96.9 | 90.85/**91.10** | 91.7/88.2 | **89.5** | 94.8 | 83.9 | 69.4 | 96.6 |
| R3F | **91.1/91.3** | 92.4/89.9 | 88.5 | 95.3 | **91.6** | **71.2** | 97.0 | **91.10/91.10** | 92.1/**88.4** | 88.4 | **95.1** | **91.2** | **70.6** | 96.2 |
| R4F | 90.1/90.8 | 92.5/89.9 | 88.8 | 95.1 | 90.9 | 70.6 | **97.1** | 90.0/90.6 | 91.8/88.2 | 88.3 | 94.8 | 90.1 | 70.1 | **96.8** |

Table 2: We present our best results on the GLUE development set for various fine-tuning methods applied to the RoBERTa Large model. On the left side table, we present our best numbers and numbers published in other papers. On the right side, we present median numbers from 10 runs for the mentioned methods.

### XNLI

We hypothesize that staying closer to the original representations is especially crucial for cross-lingual tasks, especially in the zero-shot fashion where drifting away from pre-trained representations for a single language might manifest in loss of cross-lingual capabilities. In particular, we take a look at the popular XNLI benchmark, containing 15 languages (Conneau et al., 2018). We compare our method against the standard trained XLM-R model in the zero-shot setting (Conneau et al., 2019).

| Model | en | fr | es | de | el | bg | ru | tr | ar | vi | th | zh | hi | sw | ur | Avg |
|---|---|---|---|---|---|---|---|---|---|---|---|---|---|---|---|---|
| XLM-R Base | 85.8 | 79.7 | 80.7 | 78.7 | 77.5 | 79.6 | 78.1 | 74.2 | 73.8 | 76.5 | 74.6 | 76.7 | 72.4 | 66.5 | 68.3 | 76.2 |
| XLM-R Large | 89.1 | 84.1 | 85.1 | 83.9 | 82.9 | 84.0 | 81.2 | 79.6 | 79.8 | 80.8 | 78.1 | 80.2 | 76.9 | **73.9** | 73.8 | 80.9 |
| + R3F | 89.4 | 84.2 | 85.1 | 83.7 | 83.6 | 84.6 | 82.3 | **80.7** | **80.6** | 81.1 | **79.4** | 80.1 | 77.3 | 72.6 | 74.2 | 81.2 |
| + R4F | **89.6** | **84.7** | **85.2** | **84.2** | 83.6 | 84.6 | **82.5** | 80.3 | 80.5 | 80.9 | 79.2 | **80.6** | **78.2** | 72.7 | **73.9** | **81.4** |
| InfoXLM | 89.7 | 84.5 | 85.5 | 84.1 | 83.4 | 84.2 | 81.3 | 80.9 | 80.4 | 80.8 | 78.9 | 80.9 | 77.9 | 74.8 | 73.7 | 81.4 |

Table 3: To remain consistent with prior experiments, we report an average of 5 runs of zero-shots results on the XNLI test set for our method applied to XLM-R Large. Various versions of our method win over the majority of languages. The bottom row shows the current SOTA on XNLI, which requires the pre-training of a novel model.

We present our result in Table 3. R3F and R4F dominate standard pre-training on 14 out of the 15 languages in the XNLI task. R4F improves over the best known XLM-R XNLI results reaching SOTA with an average language score of 81.4 across five runs. The current state of the art, INFO-XLM required a novel pre-training method to reach the same numbers (Chi et al., 2020).

|  | CNN/DailyMail | Gigaword | Reddit TIFU (Long) |
|---|---|---|---|
| Random Transformer | 38.27/15.03/35.48 | 35.70/16.75/32.83 | 15.89/1.94/12.22 |
| BART | 44.16/21.28/40.90 | 39.29/20.09/35.65 | 24.19/8.12/21.31 |
| PEGASUS | 44.17/**21.47**/41.11 | 39.12/19.86/36.24 | 26.63/9.01/21.60 |
| ERNIE-GEN | 44.02/**21.17/41.26** | 39.25/ 20.25/**36.53** | - |
| ProphetNet (Old SOTA) | 44.20/21.17/**41.30** | 39.51/20.42/**36.69** | - |
| BART+R3F (New SOTA) | **44.38/21.53/41.17** | **40.45/20.69/36.56** | **30.31/10.98/24.74** |

Table 4: Our results on various summarization data-sets. We report Rouge-1, Rouge-2 and Rouge-L per element in table. Following PEGASUS, we bold the best number and numbers within 0.15 of the best.

## 3.2 SUMMARIZATION

While prior work in non-standard finetuning methods tends to focus on sentence prediction and GLUE tasks (Jiang et al., 2019; Zhu et al., 2019; Zhang et al., 2020), we look to improve abstractive summarization, due to its additional complexity and computational cost, specifically we look at three datasets: *CNN/Dailymail* (Hermann et al., 2015), *Gigaword* (Napoles et al., 2012) and *Reddit TIFU* (Kim et al., 2018).

Like most other NLP tasks, summarization recently has been dominated by the fine-tuning of large pre-trained models. For example, PEGASUS explicitly defines a pre-training objective to facilitate the learning of representations tailored to summarization tasks manifesting in state-of-the-art performance on various summarization benchmarks (Zhang et al., 2019). ProphetNet (Yan et al., 2020) improved over these numbers by introducing their own novel self-supervised task as did ERNIE-GEN (Xiao et al., 2020).

Independent of the pre-training task, standard fine-tuning on downstream tasks follows a simple formula of using a label smoothing loss while directly fine-tuning the whole model without adding any new parameters. We propose the addition of the R3F term directly to the label smoothing loss. We note that R4F cannot be applied directly to generation tasks due to its reparameterization nature.

We present our results in Table 4. Our method (R3F) outperforms standard fine-tuning across the board for three tasks across all of the ROUGE metric variants. Notably, we improve *Gigaword* and *Reddit TIFU* ROUGE-1 scores by a point and four points, respectively.

## 4 REPRESENTATIONAL COLLAPSE

Catastrophic forgetting, proposed initially as catastrophic interference, is a phenomenon that occurs during sequential training where new updates interfere catastrophically with previous updates manifesting in forgetting of particular examples for a fixed task (McCloskey & Cohen, 1989). Catastrophic forgetting has been historically associated with continuous learning, and recent work (Mosbach et al., 2020) showed that catastrophic forgetting concerning the original MLM objective is not detrimental for end task training. Instead, the issue lies in optimization. Inspired by this work, we explore the related problem of representational collapse, **the degradation of generalizable representations of pre-trained models during the fine-tuning stage.** This definition is independent of a specific fine-tuning task but is rather over the internal representations generalizability over a large union of tasks. Another view of this phenomenon is that fine-tuning collapses the wide range of information available in the representations into a smaller set needed only for the immediate task and particular training set.

Measuring such degradations is non-trivial. Simple metrics such as the distance between pre-trained representations and fine-tuned representations are not sufficient (e.g., adding a constant to the pre-trained representations will not change representation power, but will change distances). One approach would be to estimate mutual information of representations across tasks before and after fine-tuning, but the estimation of mutual information is notoriously hard, especially in high-dimensions (Tschannen et al., 2019). We instead propose a series of probing experiments meant to provide us

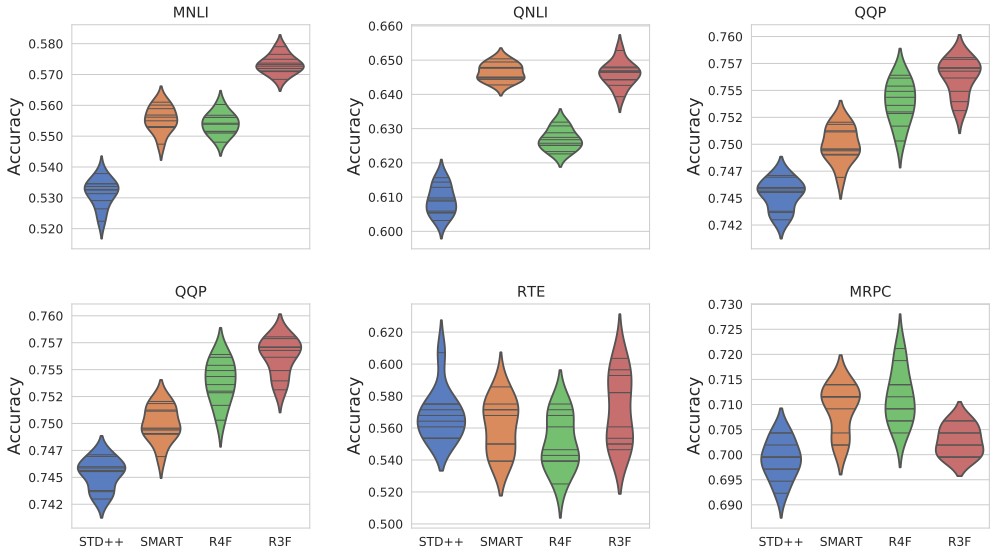

Figure 3: Results from our probing experiments comparing our proposed algorithms R3F, R4F to standard fine-tuning. Variants of our method consistently outperform past work.

with empirical evidence of the existence of representation collapse on the GLUE benchmark (Wang et al., 2018).

## 4.1 PROBING EXPERIMENTS

### PROBING GENERALIZATION OF FINE-TUNED REPRESENTATIONS

To measure the generalization properties of various fine-tuning methodologies, we follow probing methodology by first freezing the representations from the model trained on one task and then fine-tuning a linear layer on top of the model for another task. Doing this form of probing can directly measure the quality of representations learned by various fine-tuning methods and how much they collapse when fine-tuned on a sequence of tasks.

In particular, we fine-tune a RoBERTa model on SST-2 and train a linear layer for six other GLUE tasks, respectively. Our results are shown in Figure 3. Appendix A.2 presents the hyperparameters. Across all tasks, one of the two variants of our method performed best across various fine-tuning methods.

Conversely, standard fine-tuning produced representations that were worse than other fine-tuning methods across the board, hinting at the sub-optimality of standard fine-tuning. Furthermore, R3F/R4F consistently outperforms the adversarial fine-tuning method SMART.

### PROBING REPRESENTATION DEGRADATION

To show the effect of representation collapse, we propose an experiment to measure how the fine-tuning process degrades representations by sequentially training on a series of GLUE tasks. We arbitrarily select 3 GLUE tasks (QNLI, QQP, and RTE) and a source task (SST-2). We begin by training a model on our source task and then train on QNLI, QQP, and RTE in a sequential order using the best checkpoint from

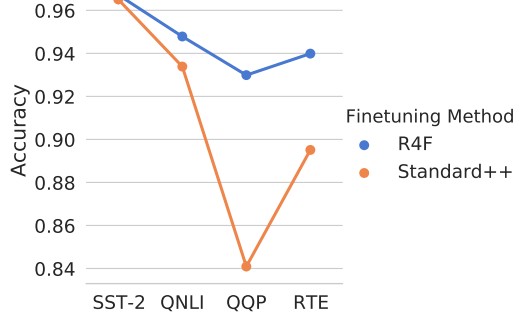

Figure 4: We show the results of the chained probing experiments. We do not show the distributional properties of the runs because there was minimal variance in the results.

the prior iteration. At each point in the chain, we probe the source task and measure performance. We compare standard SGD with the best trust-region fine-tuning approach (R4F). Our results are depicted in Figure 4.

As we can see with the standard fine-tuning process, our model diverges from the source task resulting in lower performance probes; however, with our method, the probes change much less with sequential probing resulting in better probing and end performance.

PROBING REPRESENTATION RETENTION

To further understand representational collapse's impact, we extend our probing experiments to train a cyclic chain of tasks. We showed that traditional fine-tuning degrades representations during the fine-tuning process in our prior experiments, meaning standard fine-tuning learns poorer representation compared to alternative fine-tuning methods. The dual to looking at degradation is to look at the retainment of learned representations. To do this, we take a look at cyclic sequential probing. Sequential probing involves training a model on task A, probing B, then training model fine-tuned on B and probing task C, and so forth. We then create a cyclic chain $\underbrace{A \to B \to C}_{\text{Cycle 1}} \to \underbrace{A \to B \to C}_{\text{Cycle 2}}$

from where we compare tasks via their probe performance at each cycle.

We expect probing performance to increase at every cycle; since every cycle, the task we are probing on will undergo a full fine-tuning. What we are interested in is the level of retention in representations after the fine-tuning. Specifically, we hypothesize that our method, specifically R4F, will retain representations significantly better than the Standard++ fine-tuning method.

In our experiments we consider the following sequence of GLUE tasks: SST-2 $\to$ QNLI $\to$ QQP $\to$ RTE. We defer hyperparameter values to Appendix (Section A.2).

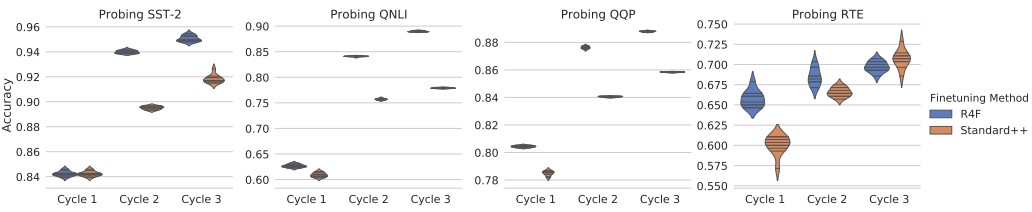

Figure 5: We present the results of cyclical sequential probing for 3 cycles.

Looking at Figure 5, we see that R4F retains the quality of representations significantly better than standard fine-tuning methods.

## 5 CONCLUSION

We propose a family of new fine-tuning approaches for pre-trained representations based on trust-region theory: R3F and R4F. Our methods are more computationally efficient and outperform prior work in fine-tuning via adversarial learning (Jiang et al., 2019; Zhu et al., 2019). We show that this is due to a new phenomenon during fine-tuning: representational collapse, where representations learned during fine-tuning degrade, leading to worse generalization. Our analysis shows that standard fine-tuning is sub-optimal when it comes to learning generalizable representations, and instead, our methods retain representation generalizability and improve end task performance.

With our method, we improve upon monolingual and multilingual sentence prediction tasks as well as generation tasks compared to standard and adversarial fine-tuning methods. Notably, we set state of the art on DailyMail/CNN, Gigaword, Reddit TIFU, improve the best-known results on fine-tuning RoBERTa on GLUE, and reach state of the art on zero-shot XNLI without the need for any new pre-training method.

We note there are many flavors of RXF that can occur with various noise distributions or perturbation strategies. We believe a larger, more general framework exists which connects trust region methods and fine-tuning in general. We leave this area of exploration for future work.

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

# A APPENDIX

## A.1 CONTROLLING CHANGE OF REPRESENTATION VIA CHANGE OF VARIABLE

Let us say we have random variables in some type of markovian chain $x, y, z; y = f(x; \theta_f), z = g(y; \theta_g)$

The change of variable formulation for probability densities is

$$p(f(x; \theta_f)) = p(g(f(x; \theta_f))) \left| \det \frac{dg(f(x; \theta_f))}{df(x; \theta_f)} \right| \tag{6}$$

Direct application of change of variable gives us

$$KL(p(f(x; \theta_f))||p(f(x; \theta_f + \Delta\theta_f))) = \tag{7}$$

$$\sum p(f(x; \theta_f)) \log \frac{p(f(x; \theta_f))}{p(f(x; \theta_f + \Delta\theta_f))} = \tag{8}$$

$$\sum p(g(f(x; \theta_f))) \left| \det \frac{dg(f(x; \theta_f))}{df(x; \theta_f)} \right| [ \tag{9}$$

$$\log p(g(f(x; \theta_f))) + \log \left| \det \frac{dg(f(x; \theta_f))}{df(x; \theta_f)} \right| \tag{10}$$

$$- \log p(g(f(x; \Delta\theta_f))) - \log \left| \det \frac{dg(f(x; \Delta\theta_f))}{df(x; \Delta\theta_f)} \right| \tag{11}$$

$$] \tag{12}$$

Let us make some more assumptions. Let $g(y) = Wy$ where the spectral norm of $W, \rho(W) = 1$. We can then trivially bound $\det W \le 1$. Then we have

$$= \sum p(g(f(x; \theta_f))) \left| \det \frac{dg(f(x; \theta_f))}{df(x; \theta_f)} \right| [\log p(g(f(x; \theta_f))) - \log p(g(f(x; \Delta\theta_f)))] \tag{13}$$

$$= \sum p(g(f(x; \theta_f))) \left| \det \frac{dg(f(x; \theta_f))}{df(x; \theta_f)} \right| \log \frac{p(g(f(x; \theta_f)))}{p(g(f(x; \Delta\theta_f)))} \tag{14}$$

$$\le \sum p(g(f(x; \theta_f))) \log \frac{p(g(f(x; \theta_f)))}{p(g(f(x; \Delta\theta_f)))} \tag{15}$$

$$= KL(p(g(f(x; \theta_f)))||p(g(f(x; \Delta\theta_f)))) \tag{16}$$

We also see that tightness is controlled by $|\det W|$, which is bounded by the singular value giving us intuition to the importance of using spectral normalization.

## A.2 EXPERIMENT HYPER-PARAMETERS

For our GLUE related experiments, both full fine-tuning and probing, the following parameters are used. For probing experiments, the difference is our RoBERTa encoder is frozen, and the encoder dropout is removed.

| Hyper Parameter | MNLI | QNLI | QQP | SST-2 | RTE | MRPC | CoLA |
|---|---|---|---|---|---|---|---|
| Learning Rate | 5e-6 | 5e-6 | 5e-6 | 5e-6 | 1e-5 | 1e-5 | 1e-5 |
| Max Updates | 123873 | 33112 | 113272 | 20935 | 3120 | 2296 | 5336 |
| Max Sentences | 8 | 8 | 32 | 32 | 8 | 16 | 16 |

Table 5: Task specific hyper parameters for GLUE experiments

| Hyper parameter | Value |
|---|---|
| Optimizer | Adam |
| Adam-betas | (0.9, 0.98) |
| Adam-eps | 1e-6 |
| LR Scheduler | polynomial decay |
| Dropout | 0.1 |
| Weight Decay | 0.01 |
| Warmup Updates | 0.06 * max updates |

| Hyper parameter | Value |
|---|---|
| $\lambda$ | [0.1, 0.5, 1.0, 5.0] |
| Noise Types | [$\mathcal{U}, \mathcal{N}$] |
| $\sigma$ | $1e-5$ |

Table 6: Hyper parameters for R3F and R4F experiments on GLUE

| Hyper Parameter | CNN/Dailymail | Gigaword | Reddit TIFU |
|---|---|---|---|
| Max Tokens | 1024 | 2048 | 2048 |
| Total updates | 80000 | 200000 | 200000 |
| Warmup Updates | 1000 | 5000 | 5000 |

Table 7: Task specific hyper parameters for Summarization experiments.

| Hyper parameter | Value |
|---|---|
| Optimizer | Adam |
| Adam-betas | (0.9, 0.98) |
| Adam-eps | 1e-8 |
| LR Scheduler | polynomial decay |
| Learning Rate | 3e-05 |

| Hyper parameter | Value |
|---|---|
| $\lambda$ | [0.001, 0.01, 0.1] |
| Noise Types | [$\mathcal{U}, \mathcal{N}$] |
| $\sigma$ | $1e-5$ |
| Dropout | 0.1 |
| Weight Decay | 0.01 |
| Clip Norm | 0.1 |

Table 8: Hyper parameters for R3F and R4F experiments on Summarization experiments.

| Hyper parameter | Value |
|---|---|
| Optimizer | Adam |
| Adam-betas | (0.9, 0.98) |
| Adam-eps | 1e-8 |
| LR Scheduler | polynomial decay |
| Learning Rate | 3e-05 |
| Dropout | 0.1 |
| Weight Decay | 0.01 |

| Hyper parameter | Value |
|---|---|
| $\lambda$ | [0.5, 1, 3, 5] |
| Noise Types | [$\mathcal{U}, \mathcal{N}$] |
| $\sigma$ | $1e-5$ |
| Total Updates | 450000 |
| Max Positions | 512 |
| Max Tokens | 4400 |
| Max Sentences | 8 |

Table 9: Hyper parameters for R3F and R4F experiments on XNLI.

