# OpenReview forum: "Better Fine-Tuning by Reducing Representational Collapse"
_ICLR.cc/2021/Conference — ICLR 2021 Poster_

### Official Review · AnonReviewer1 · 2020-10-26

**Rating:** 6
**Confidence:** 4

**Review:**


#### Summary

- This paper presents a simple but effective method rooted in trust region theory for fine-tuning pre-trained models without 'representational collapse'. Compared to previous methods (such as SMART by Jiang et al. (2019)), the newly proposed methods (R3F and R4F) are computationally simple while achieving more strong performance on several NLP tasks including GLUE, XNLI and summarization. The authors also introduce the concept of 'representational collapse', which means the degradation of generalizable representations of pre-trained models during the fine-tuning stage. Moreover, they empirically demonstrated that SMART and their proposed methods are effective in relieving representational collapse, compared to typical fine-tuning based on normal gradient descent (i.e., one without constraints).

#### Pros (Reasons to Accept)

- The paper is clearly written.
- The introduction of simple but effective & efficient methods for fine-tuning pre-trained models.
- Extensive experiments. It's good to see experiments on XNLI and summarization tasks in addition to one on GLUE.
- Strong emprical results, achieving SOTA on several tasks.

#### Cons (Reasons to Reject)

- The proposed methods (R3F and R4F) are a simple and incremental revision of the exitsing method (SMART), and there are no fundamental grounds or intuitions from which the proposed formulation is derived and justified (except for its empirical superiority).
- There is no details or explanations about how the proposed methods are directly related to trust region theory (and what exactly trust region theory is).

#### Comments

- I'm just wondering whether the proposed methods can also bring improvement to InfoXLM (in addition to XLM-R) in the XNLI experiment. If possible, showing this would make your claim much stronger.
- Why should **fine-tuned** representations be also **generalizable** in cases where we only consider the end performance of a specific target task (and the proposed methods do not bring significant improvement on the target task performance)? I understand that the proposed methods are desirable in the case of XNLI where zero-shot cross-lingual transfer explicitly requires fine-tuned representations to be still general enough to be properly transferred to other languages. Is there any other plausible story where generalizability is very important when fine-tuning for **only** a designated task?
- It would be much better if the definition of 'representational collapse' can be well-defined in a mathematical and measurable manner, instead of just relying on empirically showing its existence with probing (though it is also desirable).

---

> ### Author Response · Authors · 2020-11-16
> **Re: AnonReviewer1**
>
> Thank you for your detailed review of our paper! We appreciate and would like to address each of your comments individually.
>
> **The proposed methods (R3F and R4F) are a simple and incremental revision of the existing method (SMART), and there are no fundamental grounds or intuitions from which the proposed formulation is derived and justified (except for its empirical superiority).**
> We believe the simplicity of RXF is a benefit that manifests itself in faster compute times. Furthermore, this will allow for broader adoption for RXF than more complex/computationally expensive methods like SMART. We'd also like to reiterate another novelty of the paper: experimentation and exploration into the proposed representation collapse phenomena.
>
> **There is no details or explanations about how the proposed methods are directly related to trust region theory (and what exactly trust region theory is).**
>
> In sections 2 and 2.1, we make a connection between the SMART objective and natural gradients, which is a subfield of trust region based methods. In particular, eq 1 formally shows the relation between constrained optimization and trust region. We apologize for the confusion and will update the section to make the connection more explicit.
>
>
> **InfoXLM**
> During the paper's writing, InfoXLM was not made public; thus, we could not experiment with it.
>
> **Why should fine-tuned representations be also generalizable in cases where we only consider the end performance of a specific target task?**
> Maintaining generalizable representations even for end tasks can be seen as a form of regularization. It's trivial for models to exploit dataset-specific features to achieve excellent training performance, yet empirically we show that maintaining generalizable representations improves the generalization gap. The exact mechanism as to why this is the case is still unknown. It requires formal definitions of generalizable representations, which to our knowledge, is still an active field of research.
>
> **It would be much better if the definition of 'representational collapse' can be well-defined in a mathematical and measurable manner, instead of just relying on empirically showing its existence with probing (though it is also desirable).**
>
> We agree that a mathematical definition of representational collapse would be of great benefit, however, in order to define representational collapse, we need to define generalizable representations. This is an active area of work, hence we defer to probing experiments to measure generalization.

---

### Official Review · AnonReviewer4 · 2020-10-28
**Addressing an important problem but hard to read and understand**

**Rating:** 7
**Confidence:** 3

**Review:**

This paper proposes a method for fine-tuning to address the issue of representation collapse. The authors claim that their proposed approach is more robust to hyper-parameters of the fine-tuning process and more computationally efficient compared to its counter parts, e.g., SMART. The proposed method is called Robust Representations through Regularized Finetuning (and an extension called Robust Representations through Regularized and Reparameterized Finetuning)  the main idea, as I understand, is to minimize the amount of change in representations of the model at each training step during the fine-tuning.

In order to show that the representational collapse problem exist when using standard fine-tuning techniques, and that their method, indeed, resolves this problem, they design a series of probing experiments where the apply fine-tuning on a set of datasets/tasks in a sequential order using the best checkpoint from the prior iteration.  They measure the performance of the iteratively fine-tuned model on the source task at each step and show that with their method, the performances drops much less with sequential probing in contrast to the standard fine-tuning approach. Furthermore, they apply this iterative fine-tuning in cycles, revising the sequence of tasks in each cycle, and they find that in most cases, they get much bigger improvements in the performance of the model on the target tasks in each cycle (again compared to standard fine-tuning).

I find the problem addressed in the paper is super important and proposed solution very intuitive. I think the paper can be written in a way more clear way with a bit bigger audience in mind. In addition, in order to better show the merits of the proposed approach, if applicable maybe it would be more fair if the approach is compared with more simpler solutions to pose constraints on the amount of change in the representations, e.g., to simply  mix the data from source and target or add a distance term to the fine-tuning loss, or existing approaches like elastic weight consolidation that are used in continual learning setups (where the constraint is on the parameters of the model rather than the representations).

---

> ### Author Response · Authors · 2020-11-16
> **Re: AnonReviewer4**
>
> Thank you for taking the time to review our paper! We’re glad you found this paper to be addressing an important problem and our solution to be intuitive! We agree this paper could have been written to focus on a broader audience (i.e., fine tuning for domains other than language). Still, we wanted to explicitly focus on language fine-tuning due to various factors, including compute budget. We leave the analysis of RXF to other modalities than language as future work.
>
> For our comparisons, we wanted to compare to other various optimization techniques (FreeLB, SMART, SGD++). We agree that a more in-depth analysis would require looking at multiple approaches utilizing methods such as data-augmentations, but we believe the baselines we chose are mostly fair.
>
> Please let us know if you have any points which could make the paper clearer, we would appreciate any feedback on making this paper more accessible!

---

> > ### Comment · AnonReviewer4 · 2020-11-21
> > **Thank you for your response**
> >
> > Thank you very much for your response.
> >
> > I just wanted to clarify that what I meant by "(writing the paper with a bit bigger audience in mind" I didn't mean to try it on other domains, I think it is perfectly fine for this work to focus on the language tasks in the experiments. My point was more about the language of the paper. I consider my self as someone who is very interested in your work and would definitely want to use it or refer to it in my future work, but I wasn't familiar with things like the trust region theory, and you use this term quite often in the paper. It would have helped me a lot for example, if you simply explained what trust region theory is and why it is relevant in this context early in the paper.

---

> > > ### Author Response · Authors · 2020-11-24
> > > **Re: AnonReviewer4**
> > >
> > > Thank you for your clarification! We have updated the paper (Section~2) to explain and connect trust-region methods more explicitly with our formulation of RXF. Thank you once again for your review!

---

### Official Review · AnonReviewer3 · 2020-10-28
**A well-written paper with interesting ideas.**

**Rating:** 6
**Confidence:** 4

**Review:**

This paper presents a lightweight fine-tuning strategy motivated by the trust-region theory, which achieved SOTA results on GLUE and  XNLI using no novel pertaining approaches. The paper also introduces a new analysis by defining a notion of representational collapse and provides a new methodology for measuring it during fine-tuning, which is interesting.

In my opinion, the defining and analysis of representational collapse is the major contribution of this paper. This paper  is well-written and strongly motivated with solid experimental results.

Strength:

+ A novel view of representation learning, a.k.a. representational collapse with comprehensive evaluations and detailed analysis. It may motivate other works of robust representation learning, and it is well suited for ICLR.
+ A novel fine-tuning method which does not require extra backward computations and empirically works as well as or better than SMART
+ SOTA evaluation results on both NLU and NLG datasets

Weakness:

- The overall design of R4F is simple, as leveraging the Spectral Normalization to make the function 1-Lipschitz is not new,
- Some notions and symbols are missing, such as x~ in equation 2. And the analysis of the relationship to SMART and FreeLB is also a bit vague. I recommend the authors to carefully revise this part.

Questions:

what is the major difference between catastrophic forgetting and representational collapse?

Wha will happen if fine-tuned with different kinds of noise z? or with different sigma?

---

> ### Author Response · Authors · 2020-11-16
> **Re: AnonReviewer3**
>
> Thank you for taking the time to review our paper! We would like to address each of your concerns individually below:
>
> **The overall design of R4F is simple, as leveraging the Spectral Normalization to make the function 1-Lipschitz is not new**
> We view R4F as an extension to R3F to further bound representational collapse. We agree that the mathematics is not novel; however, we believe the application of R3F and spectral normalization in the context of finetuning and the application to representation collapse to be the novelty of R4F.
>
> **Some notions and symbols are missing, such as x~ in equation 2.**
> Thank you for your feedback; we will update our latest manuscript to fix mathematical and notational errors.
>
> **The analysis of the relationship to SMART and FreeLB is also a bit vague**
> We apologize for the confusion, At the beginning of section 2, we state the SMART objective (equation 2) and contrast it to our proposed R3F/R4F objectives (equation 3 and 4), and in section 2.1, we introduce the computation cost of other state of the art finetuning approaches. We will update our manuscript to define FreeLB explicitly and better connect sections 2 and 2.1.
>
> **What is the major difference between catastrophic forgetting and representational collapse?**
> Catastrophic forgetting has been historically associated with continuous learning, recent work (Mosbach et al. 2020 https://arxiv.org/abs/2006.04884) showed that catastrophic forgetting concerning the original MLM objective is not essential for end task training, and instead the issue lies in optimization. We introduce the phenomenon of representational collapse, which is for generalizing end task representations rather than the pre-training representations. We will update the paper to make the distinction between catastrophic forgetting and representational collapse more clear.
>
> **What will happen if fine-tuned with different kinds of noise z? or with different sigma?**
>
> We note there are many flavors of R3F that can occur with various noise distributions or perturbation strategies. We believe a larger, more general framework exists which connects trust region methods and consistency learning in general. We leave this area of exploration for future work and instead focus on a small but essential subset of trust region via R3F that shows significant improvements on several finetuning tasks across the language modality.

---

### Official Review · AnonReviewer2 · 2020-10-28
**A simple and effective method for preserving generalizability of a model when fine-tuning.**

**Rating:** 6
**Confidence:** 3

**Review:**

Summary

The paper proposes a method for finetuning pre-trained models that ensures the generalization ability of the representation is maintained. The key innovation is that the computationally expensive ascent step in the mirror descent method of SMART can be replaced by simply injecting noise. The results support the hypothesis that this works well for keeping the generalization-ability of the model. The authors also define the degradation of the generalizability of the representation during finetuning as “representational collapse”.

Strengths
- The proposed approach is based on the change of the model in the output space g.f which seems like a very sensible way to constrain the model. The proposed approach therefore shares the advantage that the “change” being minimised has some meaningful interpretation. This is in contrast to many continual learning approaches which operate purely in weight space.

- Constraining the output function g to be 1-Lipschitz is also sensible and well explained in the paper as it ensures the Bregmann-divergance-based smoothness constraint applied on the output will also constrain the representation, f.

- The experiments are quite strong. The method has been evaluated on a large range of NLP tasks using various transformers as the base model. All experiments include multiple runs and the average/median statistics have been reported.

- The approach is much faster than the closest existing method, SMART and achieves comparable accuracy in most cases.

- Overall the paper is very well written and easy to understand. The proposed novelty compared to the closest existing approach is clearly highlighted and validated by the experiments.

Concerns
- The generalization experiments in Figure 4 only compares the proposed method to standard fine-tuning with best practices (i.e. Standard++), why has a more sophisticated methods like SMART not been included in this figure? Also, the authors state that “R3F/R4F consistently outperforms the adversarial fine-tuning method SMART”, but from Figure 3 it seems that the converse is also true - in at least 2/6 of the tasks, SMART outperforms all the variants of the proposed method and is on par in two others.

- There is quite a range of performance between the variants R3F and R4F, but there aren’t any guidelines or suggestions on why this is the case or which one should be used in a particular situation.

- The results in Table 3 and Table 2 show fractional improvements over the existing methods, however not variance is reported for these numbers. Another issue is that Table 2 uses median whereas Table 3 uses average. Is there a reason for this discrepancy?

- The need for a new term “representational collapse” is not really justified in the paper. Most authors just use the term generalization. What exactly is the difference between “representational collapse” and just saying the models lacks the ability to generalize?

- Perhaps the most significant weakness of the paper is that the novelty seems a bit limited. The difference compared to SMART is not really justified in a theoretical or principled manner. For example, what are the implications for using noise samples in Eqn. 4? Is it simply a heuristic to encourage smoothness? It would be good if the authors could explain this in more detail in the paper. At the moment it just appears as if ad-hoc modifications have been made to the cost function.

Minor comments
- There are some very minor typos throughout the paper that can be fixed. Eg. “even great degree”

---

> ### Author Response · Authors · 2020-11-16
> **Re: AnonReviewer2**
>
> Thank you for your detailed review of our paper. We're glad you found our paper well written and appreciated our extensive experiments! We'd like to address your concerns.
> - We didn't include SMART in the generalization experiments mostly due to the computational cost and time constraints and the fact that SMART performed comparably to Standard++ in the early stages of the generalization experiments. We will state that SMART performance was similar to Standard++ explicitly in the paper.
> - We spent quite a bit of time figuring out what settings R3F outperformed and R4F and vice versa. We couldn't find a definitive pattern or prior, which allows us to recommend one algorithm versus the other. We'll also state this explicitly in the paper.
> - The reason for the mean/median discrepancy between Table 3/Table 2 is due to prior work. We created Table 2 from scratch and therefore decided to select the median due to it being a more robust measure. On the other hand, for Table 3, XLM-R and InfoXLM reported averages of 5 runs; therefore, to have an equivalent comparison, we also reported averages.
> - We introduced representational collapse to describe the lack of ability to generalize specifically post fine-tuning pre-trained representations. Specifically, we argue that pre-trained representations degrade during the fine-tuning stage. Similarly, catastrophic forgetting argues for degradations of generalizable representations in the continuous learning setting. We believe this term's introduction is justified due to the explorations of generalizable representations within this novel setting.
> -We believe our paper's novelty is two-fold, the proposal of RXF as a general method for fine-tuning pre-trained language representations and the exploration into this new phenomenon of representational collapse. The contribution of removing gradient ascent steps for random noise was explicitly made for decreasing SMART's computational complexity. Furthermore, the R4F extension with 1-Lip classification heads was for directly constraining the change in representations. We'll improve this section in the paper to layout our motivations more clearly. The second novelty is the exploration surrounding representational collapse.

---

### Decision · Program_Chairs · 2021-01-07
**Final Decision**

**Decision:**

Accept (Poster)

**Comment:**

This paper introduces a pair of related regularization-oriented techniques for fine-tuning pretrained transformer models for NLP tasks, and shows that both are more efficient and more effective than prior work in thorough experiments on a wide range of tasks. The techniques are motivated by the idea of 'representational collapse', which is defined as drops in the ability of a linear model trained on an input representation to solve tasks _other than_ the one being trained on.

Pros:
- The new method is demonstrated to be broadly efficient and effective on a wide range of tasks.

Cons:
- It's not clear why 'representational collapse' warrants a new term, or whether it's desirable in general.
- The motivations for some of the precise technical decisions behind the new methods are unclear.